# Antimicrobial Peptides Act-6 and Act 8-20 Derived from Scarabaeidae Cecropins Exhibit Differential Antifungal Activity

**DOI:** 10.3390/jof11070519

**Published:** 2025-07-12

**Authors:** Melissa Rodríguez, Lily J. Toro, Carolina Firacative, Beatriz L. Gómez, Bruno Rivas-Santiago, David Andreu, Jhon C. Castaño, German A. Téllez, Julián E. Muñoz

**Affiliations:** 1Studies in Translational Microbiology and Emerging Diseases (MICROS) Research Group, Translational Medicine Institute, School of Medicine and Health Sciences, Universidad del Rosario, Bogota 111221, Colombia; mrodriguezc63@gmail.com (M.R.); cfiracative@gmail.com (C.F.); beatriz.gomez@urosario.edu.co (B.L.G.); 2Center of Biomedical Research, Group of Molecular Immunology, Universidad del Quindío, Armenia 630001, Quindío, Colombia; ljtoros@uqvirtual.edu.co (L.J.T.); jhoncarlos@uniquindio.edu.co (J.C.C.); gatellez@uniquindio.edu.co (G.A.T.); 3Health Knowledge Management Research Group, Medical Science Faculty, Corporación Universitaria Empresarial Alexander Von Humboldt (UNIHUMBOLDT), Armenia 630004, Quindío, Colombia; 4Biomedical Research Unit-Zacatecas, Mexican Institute for Social Security-IMSS, Zacatecas 98000, Mexico; rondo_vm@yahoo.com; 5Proteomics and Protein Chemistry Unit, Department of Medicine and Life Sciences, Pompeu Fabra University, Barcelona Biomedical Research Park, Dr. Aiguader 88, 08003 Barcelona, Spain; david.andreu@upf.edu

**Keywords:** antifungal therapy, antimicrobial resistance, *Candida*, *Cryptococcus*, dung beetle, antimicrobial peptides

## Abstract

The number of fungal infections is steadily increasing, with considerable morbidity and mortality. Additionally, antifungal resistance is a growing concern, highlighting the need to develop new treatment options. One alternative is the use of antimicrobial peptides (AMPs). The aim of this study was to assess the in vitro and in vivo antifungal activity of designed short AMPs, Act-6 and Act 8-20, derived from cecropin transcripts of beetles from the family Scarabaeidae, against eight reference strains of the pathogenic yeasts *Candida* and *Cryptococcus.* We also evaluated the effect of these modified AMPs on the biofilm, morphogenesis, and cell morphology of *Candida albicans*, as well as the in vivo activity via a murine model of disseminated candidiasis. The AMPs herein analyzed exhibit differential antifungal activity against the yeasts assessed, and inhibit biofilm, hyphae, and pseudohyphae formation with morphological alterations in *C. albicans*. Moreover, the fungal load in mice treated with these AMPs significantly decreased. Altogether, our results suggest that Act-6 and Act 8-20 are promising antifungal molecules to control mycoses.

## 1. Introduction

Yeasts of the genus *Candida* are part of the human microbiota and are found mainly in the oral, digestive, and genital mucosa. However, when microorganisms overwhelm host defenses, an infection can occur. Candidiasis can present different clinical manifestations, ranging from skin and mucosal infections to systemic infections affecting different organs [1]. Although there are at least 15 different species of *Candida* that cause infections in humans, the most common are *Candida albicans*, *Candida glabrata*, *Candida tropicalis*, *Candida parapsilosis*, *Candida krusei,* and, more recently described, *Candida auris*, which has become a public health problem due to its antifungal resistance and high morbidity and mortality rates [2].

Invasive candidiasis, an infection associated with health care settings, is one of the most frequent invasive mycoses in the world, with mortality rates between 10% and 47% [3]. *Candida* species cause invasive disease mostly related to risk factors, such as long stays in intensive care units (ICU), treatment with broad-spectrum antibiotics, and to a lesser extent diabetes mellitus, hematological malignancies, transplants, cancer, and neutropenia [1,4,5,6].

Despite the availability of therapeutic options, such as echinocandins, azoles, and polyenes, invasive candidiasis remains highly prevalent, with an estimated 750,000 cases occurring yearly in the world [3]. In recent years, the increase in the number of cases caused by non-*albicans Candida* species, which can be resistant to antifungals, has been a concern for health authorities [7]. In *Candida* species, low susceptibility to different antifungals is related to the ability to produce biofilms. In addition, mutations in the genes involved in the synthesis of ergosterol, which is part of the cell membrane, have also been described, as well as overexpression of genes encoding export pumps that remove drugs from the microorganism [8]. Moreover, currently used antifungals present drawbacks with respect to their spectrum of activity, pharmacokinetic properties, and toxicity in the host [9]. Particularly in *C. albicans*, the ability to switch between different morphologies (yeast, hyphae, and pseudohyphae) is also considered as a virulence factor [10].

Cryptococcosis, the third most frequent invasive mycosis in the world, is caused by the encapsulated yeasts of the *Cryptococcus neoformans* and *Cryptococcus gattii* species complexes [11]. Affecting mostly persons living with HIV/AIDS, cryptococcosis often presents as meningoencephalitis, with a global estimate of 152,000 cases of cryptococcal meningitis occurring every year, resulting in 112,000 deaths and accounting for 19% of AIDS-related mortality [12]. Although infrequent, resistance to fluconazole in these yeasts is increasing, mainly due to long-term azole therapies and to fluconazole being used as a primary antifungal prophylaxis [13,14,15].

An emerging alternative treatment for microbial infections, including yeasts such as *Candida* and *Cryptococcus*, is the use of antimicrobial peptides (AMPs), produced by animals as part of their innate immune response, and with immunomodulatory functions that include leukocyte recruitment, chemotaxis stimulation, pro- and anti-inflammatory cytokine induction, as well as activation and differentiation of immune cell lines, among others [16]. AMPs, also found in plants and microorganisms, are generally 10–60 residues long with overall positive charge and hydrophobic and amphipathic properties [16]. Among AMP mechanisms of action, the most common is direct lysis, which causes pore formation, cell wall or cell membrane rupture, mitochondrial damage and gene material by loss, impairing nucleic acid and protein synthesis, and cell cycle disruption [17].

The study of AMPs from dung beetles is of particular interest, given that these insects use the feces of other animals for nesting and feeding; thus, they must deploy a highly efficient immune system based on the expression of diverse AMPs with varied mechanisms of action [18,19]. For instance, cecropins from the dung beetles *Dichotomius satanas* and *Onthophagus curvicornis* (Coleoptera: Scarabaeidae) have shown strong activity against Gram-negative bacteria, with low toxicity for eukaryotic cells [20]. It is also important to note that modification of naturally occurring AMPs or de novo design may lead to AMP analogues with improved features, including higher resistance to proteases released by microorganisms and host cells [21,22,23].

The aim of this study was to determine the in vitro antifungal effect of Act-6 and Act 8-20, two newly designed short AMPs derived from naturally occurring cecropins of the dung beetle *Oxysternon conspicillatum* [19], against the main causal agents of candidiasis and cryptococcosis. We also aimed to evaluate the in vitro effect of these cecropin-derived AMPs on the biofilm, morphogenesis and cell morphology of *Candida albicans*, as well as the therapeutic effect in vivo in a model of disseminated candidiasis. Our study contributes data supporting the relevance of AMPs as possible alternative treatments for invasive mycoses.

## 2. Materials and Methods

### 2.1. Fungal Strains

*C. albicans* ATCC 10231, *C. albicans* SC5314, *C. glabrata* ATCC 2001, *C. parapsilosis* ATCC 22019, *C. krusei* ATCC 6258 and *C. tropicalis* ATCC 750 were used as reference strains per species, as well as *C. neoformans* H99 and *C. gattii* H0058-I-2029. These reference strains were obtained commercially, were maintained in the laboratory of the MICROS Group at the Universidad del Rosario and were cryopreserved at −80 °C. Depending on the species, 24 to 48 h prior to experiments, yeasts were cultured on Sabouraud dextrose agar (SDA) medium (Difco, Becton Dickinson, Franklin Lakes, NJ, USA) at 37 °C.

### 2.2. Design and Synthesis of Antimicrobial Peptides (AMPs)

Peptides Act-6 and Act 8-20 were designed based on the N-terminal domain of the *O. conspicillatum* [20] dung beetle cecropin transcriptome. The initial template was Ox3-22, corresponding to residues 1-22 of Oxysterlin 3, with structural characteristics conferring it functional relevance within the broader cecropin family. Modifications of Ox3-22 included alterations in net charge, hydrophobic angle, and overall amino acid composition, aimed at enhancing antifungal activity.

Peptide synthesis was outsourced to Peptide 2.0 (Chantilly, VA, USA). The manufacturer conducted high-performance liquid chromatography–mass spectrometry (HPLC-MS) analyses, ensuring that the synthetic peptides attained a purity level of 95%. The lyophilized peptides were reconstituted in distilled water to a 5 mg/mL concentration, then aliquoted and stored at −90 °C until further use.

### 2.3. Evaluation of Hemolytic Activity

The hemolytic activity of Act-6, Ox3-22, and Act 8-20 was evaluated on human erythrocytes, as reported previously, with some modifications [24]. To this end, 4 mL of peripheral venous blood, treated with 1 mM of ethylenediaminetetraacetic acid (EDTA), was used. The erythrocytes were washed by centrifugation at 800× *g* for 10 min, followed by removal of the supernatant. The erythrocyte pellet was then suspended in 1X phosphate-buffered saline (PBS) (130 mM NaCl, 3 mM KCl, 8 mM Na_2_HPO_4_, and 1.5 mM K_2_HPO_4_, pH 7.4). The washing process was repeated three times, with intervening stabilization steps of 15 min at 37 °C.

Subsequently, a 1:25 dilution of the washed erythrocytes in 1X PBS was prepared. An amount of 22.5 µL of this diluted suspension was added per well in a 384-well polypropylene plate, then mixed in triplicate with 2.5 µL of serial dilutions of peptides (1.074–137.53 µM for Act-6, 2.25–144.13 µM for Ox3-22, and 5.445–174.24 µM for Act 8-20).

A 0.1% Triton X-100 (TC) was employed as a hemolysis control, and PBS as a blank [24]. The plate was then incubated at 37 °C for 18 h, followed by centrifugation at 800× *g* for 5 min. Subsequently, 3 µL of supernatant was collected, and absorbance was measured at 410 nm on a Take3 plate reader in a Biotek spectrophotometer (Epoch, Agilent, Santa Clara CA, USA).

The percentage of hemolysis was calculated using the following formula:(1)$$\%\;hemolysis=\left(\frac{X-\left(average\;of\;blanks\right)}{average\;hemolysis\;control-average\;of\;blanks}\right)\times 100$$
where X, blanks and hemolysis control are, respectively, the absorbances at 410 nm of each peptide concentration, PBS and 0.1% Triton X-100 samples.

### 2.4. Evaluation of Cytotoxic Activity

To further assess the cytotoxic activity of Act-6 and Act 8-20, metabolic activity was evaluated as an indicator of cell viability in human peripheral blood mononuclear cells (PBMCs), as reported previously [25]. All experiments involving the use of human cells were approved by the bioethics committee of Universidad del Quindío (protocol number 16, 17 May 2019) and were carried out following international recommendations. PBMCs were extracted from peripheral venous blood through cell sedimentation, and treated with 2 mM EDTA as an anticoagulant, diluted 1:1 in buffer B (0.85% NaCl, 10 mM Hepes – NaOH pH 7.4). Next, in a 15 mL conical centrifuge tube, 3 mL of iodixanol density barrier at 1.077g/mL was added, followed by the addition of 6 mL of diluted blood to obtain approximately 8 million cells. This sample was centrifuged at 700× *g* for 20 min at room temperature without braking. At the end of centrifugation, the cell interface was collected, diluted 1:1 in buffer B, and centrifuged again at 150× *g* for 10 min at room temperature. The cell pellet was then resuspended in RPMI medium with 1X antibiotic–antimycotic suspension (10,000 units penicillin, 10 mg streptomycin and 25 μg amphotericin B/mL) (Sigma-Aldrich, St. Louis, MO, USA). Cells were counted in a hemocytometer and added to a 384-well polypropylene plate at 100–150 × 10^3^ cells per well in a final volume of 25 µL.

Peptides were prepared in serial dilutions in RPMI medium (1.074–137.53 µM for Act-6, 2.25–144.13 µM for Ox3-22, and 5.445–174.24 for Act 8-20), and each concentration was added per well in triplicate. Subsequently, the plate was incubated for 18 h at 37 °C and 5% CO_2_. Resazurin was then added at a final concentration of 44 µM, and the plate was further incubated for 4 additional hours. Plate fluorescence was measured using a Synergy HTX fluorometer (Biotek, Winooski, VT, USA) at excitation and emission wavelengths of 565/600 nm. The percentage of metabolic activity was calculated as$$\%\;Metabolic\;activity=\left(\frac{\left(X-blank\right)}{\left(Cell\;control-blank\right)}\right)\times 100$$
where X is the fluorescence reading, blank is culture medium and Cell control is untreated cells.

From cytotoxic data a non-linear regression model was employed to determine the half-maximal inhibitory concentration (IC50) for peptides causing a decrease of metabolic activity exceeding 50%.

### 2.5. Antifungal Susceptibility Testing of AMPs Against Candida and Cryptococcus

The antifungal activity of Act-6 and Act 8-20 was determined in vitro for the six reference strains of *Candida* and the two *Cryptococcus* strains mentioned above. The yeasts were preserved at −80 °C and subsequently cultured on SDA at 37 °C for 24–48 h before each experiment to obtain ideal growth. The minimum inhibitory concentration (MIC) of each AMP was determined by the microdilution technique in liquid medium in a 96-well plate, based on the procedures of the reference manual M27M44S of the Clinical and Laboratory Standards Institute (CLSI) for yeast antifungal susceptibility assays [26]. Separately, each AMP was diluted in RPMI-1640 medium (Gibco, Life Technologies, New York, NY, USA) with 3-(N-morpholino) propanesulfonic acid (MOPs) (Merck, Darmstadt, Germany) in serial concentrations ranging from 0.39 to 50 µg/mL (0.1325–17.19 µM for Act-6; 0.108–17.42 µM for Act 8-20, 0.14–18.01 µM for Ox3-22). MIC was defined as the lowest peptide concentration capable of inhibiting fungal growth in vitro. When a lowest concentration could not be established because growth was similar in two consecutive wells, both concentration values were considered [13,27]. Fluconazole was diluted in RPMI-1640-MOPs in serial concentrations from 0.125 to 64 µg/mL. *C. parapsilosis* ATCC 22019 and *C. krusei* ATCC 6258 strains were used as controls for antifungal susceptibility against fluconazole, as indicated in the M27M44S manual [26].

The therapeutic index (TI) was determined as follows:$$Therapeutic\;Index\left(TI\right)=\frac{IC50\;or\;Maximum\;non\;toxic\;concentration}{MIC}$$

### 2.6. Activity Against Biofilm Formation by C. albicans

An inoculum of *C. albicans* ATCC 10231 was adjusted to 2 × 10^6^ cells/mL in RPMI 1640 with MOPs. Subsequently, 100 μL/well were placed in a 96-well plate together with 100 μL of the respective concentrations (0.09 to 50 μg/mL) of Act-6 or Act 8-20. In a different plate, amphotericin B (Sigma-Aldrich), at concentrations ranging from 0.125 to 64 µg/mL, was used as an inhibition control. *Candida* cells not exposed to peptides nor antifungal compounds were considered as growth control. Plates were incubated for 24 h at 37 °C to observe the inhibition in biofilm formation. After incubation, analysis of anti-biofilm effect was carried out using a reduction assay with 2,3-bis (2-methoxy-4-nitro-5- sulfophenyl)-2H- tetrazolium-5-carboxanilide (XTT) (Sigma-Aldrich) incubated for 3 h. Biofilm formation was calculated as follows:$$\%Biofilm\;formation=\left(\frac{\left(X-Blank\right)}{\left(Growth\;Control-Blank\right)}\right)\times 100$$
where X, growth control (amphotericin), and blank (no yeast, no reagents) are the respective absorbances 492 nm, as described previously [28].

### 2.7. Activity Against Morphogenesis of C. albicans

To observe the inhibition effect on *C. albicans* morphogenesis [29], 10^3^ cells/mL of *C. albicans* ATCC 10231, suspended in RPMI 1640-MOPS, were treated separately with Act-6 and Act 8-20, at 25 µg/mL and 6.25 µg/mL concentration, respectively, for 24 h at 37 °C. These concentrations, representing 50% of the MIC of each peptide as determined in Section 2.5, were used to avoid killing the yeasts. Subsequently, the cells were washed and fixed with 4% paraformaldehyde (Sigma-Aldrich) in PBS buffer for 30 min, adhered to glass coverslips previously covered with poly-l-lysine, and stained with 1 mg/mL of Calcofluor-White (Sigma-Aldrich) for 5 min at room temperature. The coverslips were then washed with distilled water and mounted in 1% n-propyl gallate solution (Sigma, Aldrich). Images were obtained using an inverted fluorescence microscope DMi8 (Leica Microsystems, Wetzlar, Germany). Untreated cells only received PBS.

### 2.8. Effect of AMPs on C. Albicans Morphology

The cell morphology of *C. albicans* ATCC 10231 after treatment with Act-6 and Act 8-20 was evaluated by transmission electron microscopy (TEM), as previously reported [30]. Briefly, yeast suspensions of 1 × 10^7^ cells/mL were treated with Act-6 (25 µg/mL) or Act 8-20 (6.25 µg/mL) for 24 h. Subsequently, the yeasts were fixed with 2.5% glutaraldehyde in 0.1 M (pH 7.2) for 2 h at room temperature. Post-fixation was carried out in 1% osmium tetroxide in a cacodylate buffer containing 1.25% potassium ferrocyanide and 5 mM CaCl_2_ for 2 h. Thereafter, cells were dehydrated with increasing concentrations of ethanol (50%, 70%, 90%, and 100%). Ultrathin sections were stained with uranyl acetate and lead citrate and observed in a JEM-1400 Plus microscope (JEOL Inc., Peabody, MS, USA). Photographs were taken with a Gatan camera (Gatan, Inc., Pleasanton, CA, USA).

### 2.9. In Vivo Model of Disseminated Candidiasis

Female BALB/c mice, 6 to 8 weeks old, were raised and housed at the Animal Facility of the Faculty of Veterinary Medicine and Zootechnics, Universidad Nacional de Colombia, Bogota. All animal experiments were approved by the Bioethics Committee of the Universidad Nacional de Colombia (Protocol number CB-FMVZ-UN-039-2021) and carried out following international recommendations. Animals were maintained under specific pathogen-free (SPF) conditions with food and water ad libitum. Disseminated candidiasis was induced by a 100 µL intravenous inoculation of 3 × 10^5^ *C. albicans* ATCC 10231 cells in PBS [31]. Groups of six mice were next treated by intraperitoneal injection of either Act-6 or Act 8-20 (4 mg/kg), or fluconazole (20 mg/kg) as a positive control. The negative (infected untreated) control group was treated with 500 μL of PBS. Treatment was performed every 24 h for seven days. On day 8, animals were euthanized and kidneys, spleen, and liver were collected and weighed.

Organs were thereafter homogenized individually by maceration in 1 to 2 mL of PBS and 100 µL of this suspension was plated and incubated on brain heart infusion (BHI) agar (Becton Dickinson, Franklin Lakes, NJ, USA) for determination of fungal burden by counting colony-forming units (CFU) per gram of tissue. The treatment groups were compared against the control group by a non-parametric ANOVA and a Kruskal–Wallis multiple comparison test.

Other parts of mice organs (kidneys, spleen, and liver) were placed in a tube with 10% formalin, to subsequently embed the organ in paraffin blocks to make histological sections of 5 µm with a microtome (Thermo Scientific - Microm HM325, Walldorf, Germany). The slides were afterwards processed with Periodic Acid-Schiff (PAS) staining and observed with optical microscopy (Zeiss, Oberkochen, Germany).

### 2.10. Statistical Analysis

Antifungal susceptibility testing and biofilm inhibition assay were performed in triplicate. Microscopy and in vivo models were performed in duplicate. Statistical comparisons to determine differences between control and treatment groups were made by analysis of variance (One-way ANOVA), followed by a Tukey–Kramer post-test. Statistical analyses were performed using GraphPad Prism version 9.0 (GraphPad Software, San Diego, CA, USA). *p*-values ˂ 0.05 indicate statistical significance.

## 3. Results

### 3.1. Act-6 and Act 8-20 Have Antifungal Activity Against Different Species of Candida and Cryptococcus

The design process chose Ox3-22 as a template and aimed to optimize antimicrobial properties against different *Candida* species, as well as against *C. neoformans* and *C. gattii*, by taking into account the specific residues and features known to contribute to efficacy. To this end, peptide Act-6 had three modifications (I15V; K18H; K23-), and Act 8-20 had five modifications (L9E; I15V; L17E; L21E; K23-), increasing cationicity and lowering hydrophobic momentum. In Table 1, the physicochemical characteristics of the template (Ox3-22) and the newly designed peptides are shown.

While all three peptides were active against the tested fungi (Table 2), Act 8-20 showed the best activity, with MIC values in the 0.39–25 μg/mL (0.108–8.71 μM) range. In addition, an IC_50_ of 159.2 μg/mL (55.47μM), calculated with PBMC, was only possible to be determined in Act 8-20 (Figure 1). For Act-6 and Ox3-22 the maximum concentration tested was used to calculate the TI.

### 3.2. Act-6 and Act 8-20 Affect Biofilm Formation by C. albicans

*C. albicans* ATCC 10231 yeasts treated with Act-6 and Act 8-20, at all tested concentrations, presented a significant decrease in biofilm metabolic activity, as determined by the XTT method, compared to untreated control cells. Interestingly, peptide Act-6 produced a decrease in the biofilm metabolic activity independently of the concentration, while Act 8-20 had a concentration-dependent activity similar to the amphotericin B control. At lower peptide concentrations a decrease in biofilm metabolic activity was observed, which decreased to almost 5% at higher concentrations (Figure 2).

### 3.3. Act-6 and Act 8-20 Have the Ability to Inhibit Morphogenesis in C. albicans

Yeasts treated with Act-6 and Act 8-20 showed morphological alterations in the formation of both hyphae and pseudohyphae compared to untreated control cells. In its normal, non-treated state, *C. albicans* has the ability to switch between morphologies, which is an important virulence factor (Figure 3).

### 3.4. Act-6 and Act 8-20 Induce Morphological Alterations in C. albicans

Act-6 and Act 8-20 at 25 and 6.25 µg/mL respectively caused important alterations in *C. albicans* yeast cells, including cell membrane invagination and contracted intracellular contents (Figure 4).

### 3.5. Fungal Load Decrease in Mice Organs Treated with AMPs

In a disseminated candidiasis infection model, both Act-6 and Act 8-20 peptides showed antifungal activity. In untreated mice, fungal burden was highest in the spleen, followed by the kidneys and liver. Treatment with Act-6 resulted in significant fungal burden reductions across all organs, exceeding even the efficacy of fluconazole. Act 8-20, for its part, only decreased fungal burden significantly in the spleen, with reductions in the liver and kidneys that were not statistically significant (Figure 5).

### 3.6. Tissues of Mice Treated with AMPs Are More Preserved

Histopathological analysis showed an important decrease in the fungal burden in the livers of mice treated with either Act-6 or Act 8-20 compared to untreated ones (Figure 6). There were no differences in the histopathological analysis of kidneys and spleens of untreated and treated mice.

## 4. Discussion

Insects respond to microbial infection by rapidly synthesizing and secreting antimicrobial peptides (AMPs) into the hemolymph, triggering pathogen-killing activities [18]. These molecules have demonstrated antimicrobial, anti-inflammatory, and, in some cases, anti-cancer activities [32]. Mammalian AMPs also possess antimicrobial capacity and immunomodulatory functions, including cell chemotaxis, cytokine induction, and cellular differentiation, promoting angiogenesis, wound healing, and infection resolution [33,34].

AMPs in the hemolymph of Coleoptera insects are of interest due to their diversity and prevalence, including attacins, tenecins, defensins, coleoptericins, and cecropins [35]. While some AMPs are naturally produced by Scarabaeidae dung beetles [19], the AMPs in this study were designed as derivatives of beetle peptides, modified in net charge, hydrophobic angle, and amino acid composition.

Act-6 and Act 8-20 showed antifungal effects, particularly against non-*albicans Candida* and *Cryptococcus* species. Act 8-20 displayed better antimicrobial activity, even against *C. albicans*, the most prevalent cause of *Candida* infections [36]. Our results highlight the potential of structurally modifying natural peptides to enhance antimicrobial capacity, as evidenced by the decreased MIC values. Others synthetic AMPs obtained by computational design (PNR20 and PNR20-1), have shown MICs between 25 µM and 100 µM, describing complete growth inhibition of *Candida* species [30]. Further research should investigate whether Act 8-20’s modifications reduce cytotoxicity and proteases susceptibility, and if its antifungal activity extends to other microorganisms, such as bacteria, mycobacteria, and molds [16,21,37].

The Act-6 and Ox3-22 peptides demonstrated low hemolytic activity, consistent with other studies on cecropin-like peptides [19,24,35]. However, the studied peptides exhibited varying cytotoxicity against mammalian cells, and Act 8-20 showed higher cytotoxicity than Ox3-22 and Act-6. This underscores the need to balance antimicrobial activity and host cell toxicity.

Act-6 and Act 8-20 reduced the metabolic activity of *C. albicans* ATCC 10231 (Figure 2), which could reduce the ability to form biofilms, a structure that protects yeasts from antifungals and the immune system [38]. In addition, biofilms have the ability to gradually release yeasts into the bloodstream, causing disseminated infections such as candidemia [39]. The ability of the studied peptides to the possible inhibition of the biofilm formation, an important virulence factor, adds to the antifungal properties of AMPs against biofilm-forming microorganisms, including fungi such as *C. albicans*. A similar effect has been reported, for instance, with peptides analogous to LL37, which were able to inhibit biofilm formation in *C. albicans* at concentrations ranging between 2.5 and 20 µM, approximately equivalent to MIC values between 3 µg/mL and 30 µg/mL [40]. AMP-17, from *Musca domestic*a, also inhibits *C. albicans* biofilm formation [41]. Further experiments are needed to evaluate the peptides’ effect on mature biofilms.

Act-6 and Act 8-20 also inhibited in *C. albicans* ATCC 10231 morphogenesis. Hyphae and pseudohyphae enable *C. albicans* to penetrate the host epithelial barrier and evade the immune system [40,42]. This morphological alteration in yeasts treated with 25 and 6.25 µg/mL of Act-6 and Act 8-20, respectively, therefore affects the pathogenicity of this *Candida* species. Compared to AMP-17 from the *Musca domestica*, which alters *C. albicans* at 40 μg/mL [41,43], Act-6 and Act 8-20 showed a stronger effect, requiring a smaller concentration to inhibit hyphae and pseudohyphae.

TEM revealed that Act-6 and Act 8-20 induce morphological changes in *C. albicans* ATCC 10231, including cell membrane condensation, abnormal vacuoles formation, and membrane disruption. Other AMPs, like PNR20 (100 µM) [30], and K40H, an immunoglobulin M-derived peptide [44], also induce morphological alterations, such as cytoplasm disintegration, vacuolation, and cell wall swelling [42]. The in vivo model using BALB/c mice showed a significant antifungal effect of Act-6 and Act 8-20 in mice infected with *C. albicans* ATCC 10231, and the animals presented lower fungal burden in the liver compared to the spleen and kidneys. The treatment with Act-6 resulted in a significant decrease in fungal burden across all organs. In contrast, Act 8-20 treatment yielded a significant difference only in the spleen. While a decrease in fungal burden was observed in the liver and kidneys with Act 8-20, this was not statistically significant, and the data showed greater dispersion. This suggests that Act-6 may have a more potent and consistent antifungal effect than Act 8-20 in this model.

The differences in peptide activity may relate to Act 8-20’s greater hydrophobicity compared to Act-6, which could reduce bioavailability or stability. Hydrophobic AMP regions can bind to serum proteins affecting their stability and free concentration [45,46], thus reducing antimicrobial efficacy.

Mouse model studies have shown that AMPs can control candidiasis, such as Gomesin, a cationic peptide of 18 amino acids obtained from the caranguejeira spider from Brazil, which protected *C. albicans*-infected mice and induced pro-inflammatory cytokines such as TNF-α, IFN-ɣ, and IL-6 [31]. Histopathological analysis also correlated with the data obtained in the fungal burden assays. The liver tissues were more preserved in the animals treated with either Act-6 or Act 8-20 compared to untreated animals. However, when performing this analysis, we did not find significant differences between the spleen and kidney samples of the animals studied. Many other AMPs have shown a promising effect in the control of invasive fungal infections. For instance, the use of ToAP2D in the control of animals with sporotrichosis caused by the dimorphic fungi *Sporothrix globosa*, has been reported [47]. Histatin-5 (Hst-5) has also shown broad antifungal activity, and recently, a new formulation of this AMP showed a decreased fungal burden in a model of oral candidiasis, as could be evidenced in the tongue histopathology of mice infected and treated with Hst-5 [48].

Our results suggest that Act-6 and Act 8-20, beetle-derived AMPs, exhibit differential antifungal activity and warrant further exploration as potential treatment options. However, a deeper investigation into the specific mechanisms of action of the peptides are needed, particularly addressed to determine how these molecules interact with the cells of diverse pathogenic yeasts, considering that among genera there are clear differences in cell wall composition, and even the presence of a polysaccharide capsule, as is the case of *Cryptococcus* species. These AMPs show promise against *Candida* and *Cryptococcus* species, major causes of life-threatening infections. Future research should investigate their mechanisms, cytotoxicity in eukaryotic cells, and potential for controlling disseminated candidiasis and other invasive fungal infections.

## Figures and Tables

**Figure 1 jof-11-00519-f001:**
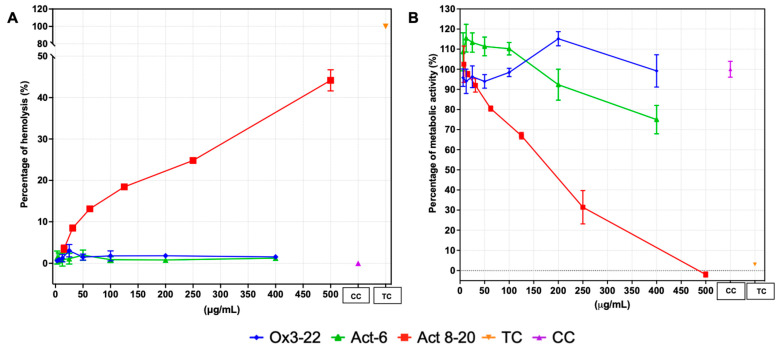
Hemolysis of Act-6, Act 8-20 and Ox3-22 in human erythrocytes and PBMC cytotoxicity. (**A**) Hemolytic activity. (**B**) PBMC cytotoxicity. TC: Triton X100 control; CC: Cell control.

**Figure 2 jof-11-00519-f002:**
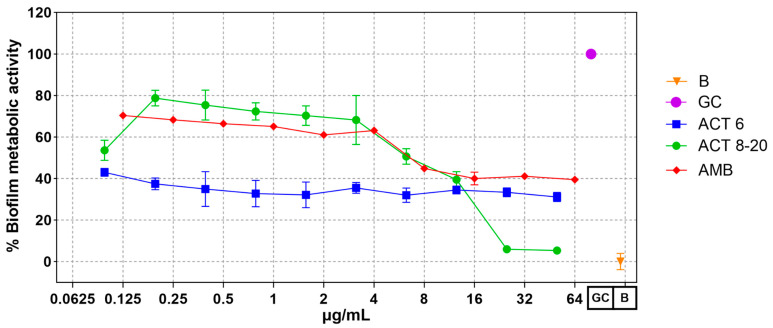
Biofilm metabolic activity of *Candida albicans* ATCC 10231 treated with antimicrobial peptides Act-6 and Act 8-20 at 0.095 to 50 μg/mL. Amphotericin B 0.125 to 64 μg/mL was used as control. The metabolic activity of *C. albican*s not exposed to peptides is indicated as a growth control (GC). The optical density of reagents was used as blank (B).

**Figure 3 jof-11-00519-f003:**
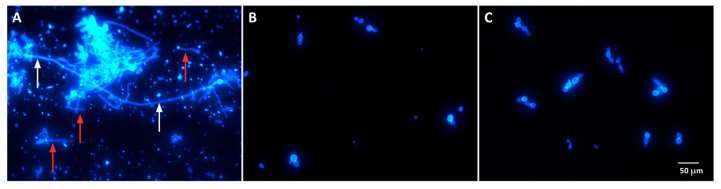
*Candida albicans* yeasts ATCC 10231 cells untreated (**A**) and treated with 25 µg/mL of Act-6 (**B**) and 6.25 µg/mL of Act 8-20 (**C**). White and red arrows indicate hyphae and pseudohyphae formation, respectively, in untreated cells. Budding blastoconidia are observed in treated cells. Images are shown at a magnification of at 40×.

**Figure 4 jof-11-00519-f004:**
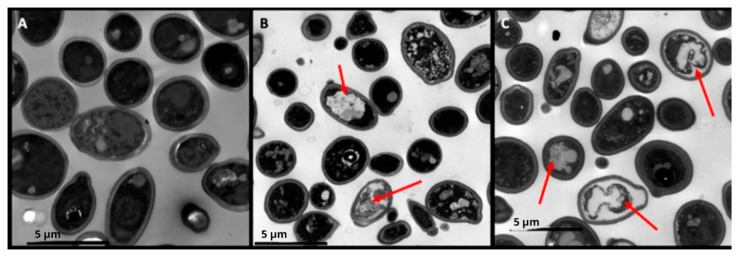
*Candida albicans* ATCC 10231 yeasts untreated (**A**), and treated separately with 25 µg/mL of Act-6 (**B**) and 6.25 µg/mL Act 8-20 (**C**). Red arrows show morphological alterations such as invagination of the cell membrane and contracted intracellular contents. Microphotographs were taken with a Gatan camera and a transmission electron microscope.

**Figure 5 jof-11-00519-f005:**
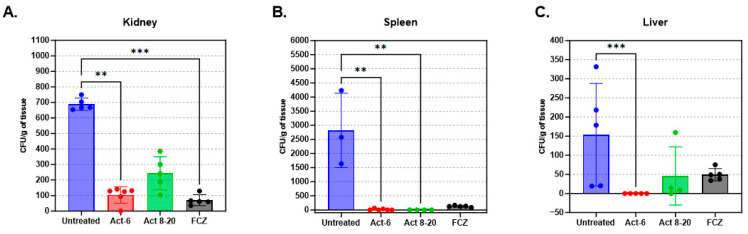
Colony-forming units (CFUs) of (**A**) kidneys, (**B**) spleens, and (**C**) livers (L) of BALB/c mice infected intravenously with 3 × 10^5^ yeasts of *Candida albicans* ATCC 10231. The animals were sacrificed after 8 days of infection. The treatment groups were compared against the untreated group by a one-way ANOVA with the Kruskal–Wallis multiple comparison test, statistical significance *p* values 0.0021 (**), 0.0002 (***).

**Figure 6 jof-11-00519-f006:**
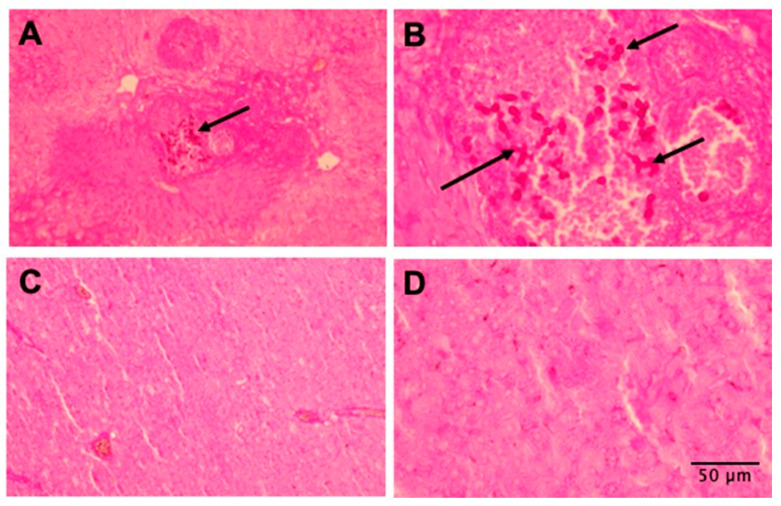
Histopathological comparison of livers of BALB/c mice with disseminated candidiasis by *Candida albicans* ATCC 10231. (**A**) Untreated animals (infected only), 10× magnification, (**B**) untreated animals (infected only), 40× magnification, (**C**) animals infected and treated with the Act 8-20 peptide, 10× magnification, (**D**) infected and treated animals with Act 8-20 peptide, 40× magnification. PAS coloring. Arrows show yeast cells invading the tissue.

**Table 1 jof-11-00519-t001:** Physicochemical characteristics of the evaluated peptides.

AMP	Sequence	Length	Molecular Weight (Da)	pI	GRAVY	A*kT/e (Polar Angle)
**Ox3-22**	GSKRWRKFEKRVKKVFEHTKEA	22	2775.26	10.64	−1.709	18.951 (113.2)
**Act-6**	GSKRWRKFEKRVKKIFEKTKEAK	23	2908.49	10.77	−1.822	16.645 (115.99)
**Act 8-20**	GSKRWRKFLKRVKKIFLHTKLAK	23	2869.59	12.32	−0.839	19.194 (107.57)

GRAVY: Grand average of hydropathicity; pI: Theoretical Isoelectric point.

**Table 2 jof-11-00519-t002:** Antifungal activity of antimicrobial peptides against different species/strains of *Candida* and *Cryptococcus.*

AMP	Values in μg/mL (μM)	IC50	*C. albicans* ATCC 10231	*C. albicans* SC5314	*C. glabrata* ATCC 2001	*C. parapsilosis* ATCC 22019	*C. krusei* ATCC 6258	*C. tropicalis* ATCC 750	*C. neoformans* H99	*C. gattii* H0058-I-2029
**Ox3-22**	MIC		>50 (18.01)	>50 (18.01)	1.56 (0.56)	0.39 (0.14)	0.78 (0.28)	25 (9)	25 (9)	1.56 (0.53)
	TI (hemolysis)	>400 (144.13)	<8	<8	>257.38	>1029.51	>514.75	>16.01	>16.01	>257.38
	TI (PBMC cytotoxicity)									
**Act-6**	MIC		>50 (>17.19)	25–50 (8.59–17.19)	3.12 (1.07)	3.12 (1.07)	1.56 (0.53)	>50 (>17.19)	25 (8.59)	1.56 (0.53)
	TI (hemolysis)	>400 (137.53)	<8	>16.01	128.53	>128.53	>259.49	<8	>16.01	>259.49
	TI (PBMC cytotoxicity									
**Act 8-20**	MIC		12.5 (4.35)	6.25–12.5 (2.17–4.35)	12.5–25 (4.35–8.71)	25 (8.71)	3.1 (1.08)	6.25 (2.17)	3.1–6.25 (1.08–2.17)	0.39 (0.108)
	TI (hemolysis)	>500 (174.24)	>40.06	40.06–80.29	20–40.06	>20	>161.33	>80.29	80.29–161.33	>1613.33
	TI (PBMC cytotoxicity)	159.2 (55.47)	12.75	12.75–25.56	6.37–12.75	6.37	51.36	25.56	25.56–51.36	513.61

The therapeutic index (TI) was calculated for the hemolytic and cytotoxic concentrations.

## Data Availability

The original contributions presented in this study are included in the article. Further inquiries can be directed to the corresponding author.

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
