# Peer review of "Antimicrobial Peptides Act-6 and Act 8-20 Derived from Scarabaeidae Cecropins Exhibit Differential Antifungal Activity"

_jof, 2025, doi:10.3390/jof11070519_

Round 1
Reviewer 1 Report (Previous Reviewer 4)
The manuscript by Rodríguez et al. fits in the line of research aiming at the development of new antimicrobials, in particular antifungals, to face the problems related to the toxicity of the drugs currently in use and the spread of antimicrobial resistance.
The work describes the activity of two peptides, derived by a previously described cecropin (Oxysterlin 3) from Oxysternon conspicillatum through specifically designed modifications.
Although the article is overall well written, it does not provide a relevant contribution to the research topic since the results presented are too preliminary. Some of the reported findings are merely descriptive and lack quantitative data. Other experiments should be performed to clarify the mechanism of action of the investigated molecules. Likely, further modifications of these peptides, in particular Act 8-20, could lead to an increased efficacy and a reduced toxicity.
see attachment

Author Response
Responses to reviewer 1
All suggestions for bibliographic references were considered and organized in the document.
Regarding the materials and methods, the grammatical and content corrections suggested by the reviewer were made.
- Ox3-22 peptide concentrations were added in the paragraphs indicated by the reviewer.
- Regarding the observation of Paragraph 2.7. Activity against morphogenesis of C. albicans: The phrase: analysis LAS X software (mentioned in line 224) was removed from the manuscript since the experiment was not quantitative as indicated by the reviewer.
The reviewer indicates that the histological analysis of organs of the infected mice (untreated controls and treated with peptides) did not include the collection of quantitative data. This is an important observation that we will take into account for future studies in order to have quantitative data in histopathological analyses.
Line 265: What is the meaning of the sentence “…in vivo model were performed in duplicate.”?
It is a way of explaining that both the animal model and the CFU analysis were performed twice during the time of the study.
Table 1 is not fully readable and must be centered.
Table 1 was corrected in the document, taking into account the reviewer's comments.
Review: The effect of the investigated peptides on C. albicans ATCC 10231 metabolic activity are not apparently related to the previously obtained MIC values. Moreover, amphotericin B, a fungicidal agent, has a lower effect even at high concentrations. Is there some possible explanation? What are the values of MIC and MFC for amphotericin B against C. albicans ATCC 10231?
R/: We agree with the reviewer's comments. However, Figure 2 shows a significant decrease in the metabolic activity of yeasts exposed to high concentrations of amphotericin B. This effect was possibly not so marked because this analysis was not performed on planktonic cells but on Candida biofilms, and it is known that biofilm hinders the penetration of antifungal drugs. The MIC value of AMB for C. albicans ATCC 10231 is 0,19 - 0,25 μg/mL. Please consider that normal antifungal susceptibility testing uses a yeast concentration of 1-5×103 cell/ml (CLSI manual), while biofilm experiments start with 2×106 cell/ml.
Reviewer 2 Report (Previous Reviewer 3)
Dear authors,
I have carefully reviewed the changes and corrections made to your manuscript. In my opinion, the revised version shows significant improvement compared to the original submission. I believe this version is ready for publication in the Journal of Fungi.
Best regards,
Dear authors,
I have carefully reviewed the changes and corrections made to your manuscript. In my opinion, the revised version shows significant improvement compared to the original submission. I believe this version is ready for publication in the Journal of Fungi.
Best regards,
Author Response
I have carefully reviewed the changes and corrections made to your manuscript. In my opinion, the revised version shows significant improvement compared to the original submission. I believe this version is ready for publication in the Journal of Fungi.
Dear reviewer 2, we appreciate your comments.
Reviewer 3 Report (New Reviewer)
The study addresses a critical need for novel antifungal agents and presents a well-structured analysis of the antifungal potential of two synthetic antimicrobial peptides (AMPs), Act-6 and Act 8-20, derived from cecropins of Scarabaeidae beetles. The authors explore their activity against Candida and Cryptococcus species, both in vitro and in vivo, and assess their effects on biofilm formation, morphogenesis, and fungal cell morphology. The study is of high relevance given the increasing incidence of fungal infections and the growing problem of antifungal resistance.
The authors employ a multi-faceted approach, including: MIC determination against multiple fungal strains, hemolytic and cytotoxicity assays, biofilm inhibition and morphogenesis studies, transmission electron microscopy for morphological analysis, and an in vivo murine model of disseminated candidiasis. The rational design of Act-6 and Act 8-20 based on Ox3-22, with modifications to improve antifungal activity and physicochemical properties, is well-executed and clearly described. The use of a murine model adds translational value to the findings, and the inclusion of fluorescence and electron microscopy, as well as histological analysis of infected tissues, provide strong visual evidence of the peptides’ effects.
This manuscript presents a well-composed investigation of the antifungal properties of two novel AMPs. However, to enhance the scientific potential of the study, the authors should consider addressing the limitations outlined below:
-The study relies solely on reference strains. Including clinical isolates, especially drug-resistant, would strengthen the clinical relevance of the findings.
-No information is provided on the peptides’ stability in the presence of serum.
-Act 8-20, while more potent, exhibits higher cytotoxicity toward PBMCs. This raises concerns about its safety.
-Although morphological changes are observed, the molecular mechanism of antifungal action remains speculative. Further studies (e.g., membrane integrity assays, gene expression profiling) are required.
-The potential for synergistic effects with existing antifungals (e.g., fluconazole) is not explored. This could be important for combination therapy.
-The study focuses on biofilm formation inhibition but does not assess the peptides’ ability to disrupt mature biofilms, which are more clinically relevant.
Author Response
Responses to reviewer 3
The study relies solely on reference strains. Including clinical isolates, especially drug-resistant, would strengthen the clinical relevance of the findings.
R/ We appreciate the reviewer's observation and fully agree. We believe it is essential to include in future studies analyses with resistant isolates of the species included in this study.
-No information is provided on the peptides’ stability in the presence of serum.
R/ It is interesting to observe the serum stability not only of the Act 6 and Act 8-20 peptides, but also of the other peptides we have in our antimicrobial peptide bank; this will undoubtedly further boost the use of these molecules as potential therapeutic agents.
-Act 8-20, while more potent, exhibits higher cytotoxicity toward PBMCs. This raises concerns about its safety.
R/ We completely agree. Toxicological studies are the next step we'll take, as it will help us verify the potential side effects of treatment with this antimicrobial peptide.
-Although morphological changes are observed, the molecular mechanism of antifungal action remains speculative. Further studies (e.g., membrane integrity assays, gene expression profiling) are required.
R/ We agree with the reviewer. Our study shows significant morphological changes in yeasts treated with the antimicrobial peptides studied. We will undoubtedly take the reviewer's valuable contributions into account in future work on this research line .
-The potential for synergistic effects with existing antifungals (e.g., fluconazole) is not explored. This could be important for combination therapy.
R/ We agree with the reviewer on the importance of synergism. However, before conducting in vitro and in vivo synergism studies, it is important to first describe the antifungal effect of the antimicrobial peptides it self to select the most promising ones and subsequently conduct synergism studies with antifungals such as fluconazole.
-The study focuses on biofilm formation inhibition but does not assess the peptides’ ability to disrupt mature biofilms, which are more clinically relevant.
R/ We are grateful for the valuable comment and agree with the reviewer; the data on the effect of these antimicrobial peptides on mature biofilm produced by C. albicans is currently being studied by our research group. We expect to publish the results soon.
Round 2
Reviewer 1 Report (Previous Reviewer 4)
Although the authors did not provide responses for all the reviewer's comments, the revised version of the manuscript is improved.
Some issues previously raised were not properly considered by the authors and must be addressed:
lines 83-84: the sentence, although modified, is not clear, may be the meaming was "...mitochondrial damage, gene material loss, impairing nucleic acid and protein synthesis, ..." if so, please correct.
line 90: Reference 20 (Téllez Ramirez, G.A.; Osorio-Méndez, J.F.; Henao Arias, D.C.; Toro S., L.J.; Franco Castrillón, J.; Rojas-Montoya, M.; Castaño Osorio, J.C. New Insect Host Defense Peptides (HDP) From Dung Beetle (Coleoptera: Scarabaeidae) Transcriptomes. J. Insect Sci. 2021, 21, 12, doi:10.1093/jisesa/ieab054.) is not the correct reference at this point. The paper previously cited in line 88 as ref. 19 should be the correct one (Henao Arias, D.C.; Toro, L.J.; Téllez Ramirez, G.A.; Osorio-Méndez, J.F.; Rodríguez-Carlos, A.; Valle, J.; Marín-Luevano, S.P.; Rivas-Santiago, B.; Andreu, D.; Castaño Osorio, J.C. Novel Antimicrobial Cecropins Derived from O. curvicornis and D. satanas Dung Beetles. Peptides 2021, 145, 170626, doi:10.1016/j.peptides.2021.170626.20). Please correct.
line 96: Reference 19 (Henao Arias, D.C.; Toro, L.J.; Téllez Ramirez, G.A.; Osorio-Méndez, J.F.; Rodríguez-Carlos, A.; Valle, J.; Marín-Luevano, S.P.; Rivas-Santiago, B.; Andreu, D.; Castaño Osorio, J.C. Novel Antimicrobial Cecropins Derived from O. curvicornis and D. satanas Dung Beetles. Peptides 2021, 145, 170626, doi:10.1016/j.peptides.2021.170626.20) is not the correct reference at this point. The paper incorrectly cited in line 90 as ref. 20 should be the correct one (Téllez Ramirez, G.A.; Osorio-Méndez, J.F.; Henao Arias, D.C.; Toro S., L.J.; Franco Castrillón, J.; Rojas-Montoya, M.; Castaño Osorio, J.C. New Insect Host Defense Peptides (HDP) From Dung Beetle (Coleoptera: Scarabaeidae) Transcriptomes. J. Insect Sci. 2021, 21, 12, doi:10.1093/jisesa/ieab054). please correct.
Lines 195-200: my previous comment was "It is stated that fluconazole was chosen as positive control, but no data have been mentioned in the relative paragraph of the Results section, so these lines can be deleted. If so, also reference 25 must be deleted and the reference numbers changed accordingly." Why only the lines 195-197 have been deleted and the cited reference (now 27) was maintained? Also lines 197-200 (and ref. 27) should be deleted.
I strongly remark my previous comment on Figure 3, panel A: "While the with arrows surely indicate hyphae, the structures indicated by red arrows (although the image is not very clear) more probably refer to germinating yeast cells (germ tubes, the initial phase of hypha formation). Pseudohyphae are likely present (but cannot be clearly individuated) within the mass of fluorescent material in the upper part of the figure." In this regard, see figure 1 panel (b) of the cited reference 10 Sudbery, P.; Gow, N.; Berman, J. The Distinct Morphogenic States of Candida albicans. Trends Microbiol. 2004, 12, 498 317–324, doi:10.1016/j.tim.2004.05.008. To avoid misunderstandings and confusion to the readership, the legend should be changed as follows: "White and red arrows indicate hyphae and hypal germ tubes formation, respectively, in untreated cells."
In lines 416, 418, and 425, the previous references (in red) [40],[41], and[42] should be obviously deleted.
Author Response
Review 1: Detailed comments for Authors
The manuscript is overall well written, but several issues should be addressed. Microorganism and insect names in the text (line 429) as well as in references should be in italics.
R/ We appreciate the reviewer´s suggestion. All species names have been italicized throughout the manuscript, including in the references. It is important to note that the species names do not appear in italics in the titles of the different sections of the manuscript due to the journal's formatting.
Specific issues:
Introduction
Line 53: a more recent paper may be added to refs 1,4, and 5 (or may substitute the older reference), e.g.: Lass-Flörl C, Kanj SS, Govender NP, Thompson GR 3rd, Ostrosky-Zeichner L, Govrins MA. Invasive candidiasis. Nat Rev Dis Primers. 2024;10(1):20. doi: 10.1038/s41572- 024-00503-3.
R/ The reference suggested by the reviewer was added to the manuscript
Line 64: Ref. 8 may be substituted by a more recent reference, e.g.: Carmo A, Rocha M, Pereirinha P, Tomé R, Costa E. Antifungals: From Pharmacokinetics to Clinical Practice. Antibiotics (Basel). 2023; 12(5):884. doi: 10.3390/antibiotics12050884.
R/ The reference suggested by the reviewer was added to the manuscript
Line 73: a more recent paper may be added to refs 12 and 13, e.g.: Melhem MSC, Leite Júnior DP, Takahashi JPF, Macioni MB, Oliveira L, de Araújo LS, Fava WS, Bonfietti LX, Paniago AMM, Venturini J, Espinel-Ingroff A. Antifungal Resistance in Cryptococcal Infections. Pathogens.
2024; 13(2):128. doi: 10.3390/pathogens13020128.
R/ The references suggested by the reviewer were added to the manuscript
Line 76: it should be “animals as part of their innate immune response…”
R/ The statement in the manuscript was corrected as suggested by the reviewer
Lines 82-83: the sentence “…mitochondrial damagegene material by loss impairing nucleic acid and protein synthesis…” doesn't make sense and needs to be rewritten.
R/ The statement in the manuscript was corrected
Line 89: Reference 18 is not the correct reference at this point. The paper previously cited in line 87 as ref. 17 should be the correct one. Maybe that the two references have been exchanged? Please check and correct.
R/ reference 18 was modified
Line 95: Reference 17 is not the correct reference at this point. The paper previously cited in line 89 as ref. 18 should be the correct one. Please check and correct.
R/ Reference 17 was corrected
Lines 96-98: It should be better to write “We also aimed to evaluate the in vitro effect of these cecropin-derived AMPs on the biofilm, morphogenesis and cell morphology of Candida albicans, as well as the therapeutic effect in vivo in a model of disseminated candidiasis.
R/ This reviewer's suggestion was considered and added to the mentioned paragraph.
Materials and Methods section
In Paragraphs 2.3, 2.4 and 2.5 the “parent” peptide Ox3-22, used in comparison with the derived peptides, should be mentioned along with its concentrations (lines 132, 161, and 186-187).
R/ Ox3-22 peptide concentrations were added in the paragraphs indicated by the reviewer.
Paragraph 2.3. Assessment of the hemolytic activity of the AMPs.
Lines 132-133: as described below for biological tests, it would be useful to have an indication of the concentrations of peptides used for the evaluation of the hemolytic activity, since the sentence “…serial dilutions of peptides covering a 170 - 200 μM concentration range.” is not a clear statement.
R/ Following the reviewer's recommendations, the concentrations used for each peptide, including the Ox3-22 peptide, were indicated more precisely.
Line 134: Reference 16 is not the correct reference at this point. It should be Ref. 22. Please check and correct.
R/ Reference 16 was reviewed and modified
Line 135: The incubation time of 18 h is very unusual. Incubation for the evaluation of the hemolytic activity of peptides is generally 30 minutes, 1 hour or 2 hours. Why this incubation time has been chosen?
R/ Thanks to the reviewer for noticing this. Indeed, we decided to incubate the plate at 37 °C for 18 h, instead of 1 hour, to better demonstrate the possible hemolytic effect of the studied peptides. Clearly, this hemolytic effect was low during the long time analyzed.
Paragraph 2.4. Evaluation of the cytotoxic activity of the AMPs. Line 161: see previous comment to Lines 132-133.
R/ Following the previous comment made by the reviewer, the concentrations used for each of the peptides were added to this section.
Lines 171-172: cell control cannot be 0.1% Triton 100X. It should be medium without peptides. Please check and correct.
R/ Sentence modified in the manuscript
Lines 174-175: It should be better to write “…for peptides causing a decrease of metabolic activity exceeding 50%.” instead of “…for peptides exceeding 50% of metabolic activity.”
R/ Sentence modified in the manuscript
Paragraph 2.5. Antifungal susceptibility testing of AMPs against Candida and Cryptococcus Lines 190-195: It is stated that fluconazole was chosen as positive control, but no data have been mentioned in the relative paragraph of the Results section, so these lines can be deleted. If so, also reference 25 must be deleted and the reference numbers changed accordingly.
R/ We appreciate the reviewer's suggestion. The modification was made into the manuscript, appearing now between lines 255 and 262.
Paragraph 2.6. Activity against biofilm formation by C. albicans
Line 211: The sentence in line 202 should be changed since, as correctly stated in Legend of Figure 2, the metabolic activity of C. albicans not exposed to peptides represents the growth control (GC), while amphotericin B is used as a positive control, i.e. as a drug active on yeast cells.
R/ Information adjusted in the manuscript as expressed in the legend of figure 2
Paragraph 2.7. Activity against morphogenesis of C. albicans
The designed experiment does not involve the collection of quantitative data (e.g. the percentage of germinating cells, pseudohyphae and/or the length of hyphae after observation of an equal number of cells in the untreated control and in the samples treated with peptides). For what purpose was used the image analysis LAS X software (mentioned in line 224)?
R/ Regarding the observation of Paragraph 2.7. Activity against morphogenesis of C. albicans: The phrase: analysis LAS X software (mentioned in line 224) was removed from the manuscript since the experiment was not quantitative as indicated by the reviewer.
Paragraph 2.9. In vivo model of disseminated candidiasis
Lines 247-248: The sentence should be “Treatment was performed every 24 h for seven days.”
R/ Sentence modified in the manuscript
The histological analysis of organs of the infected mice (untreated controls and treated with peptides) did not include the collection of quantitative data. The experimental design could be improved.
R/ The reviewer indicates that the histological analysis of organs of the infected mice (untreated controls and treated with peptides) did not include the collection of quantitative data. This is an important observation that we will consider for future studies in order to have quantitative data in histopathological analyses.
Paragraph 2.10. Statistical analysis
Line 265: What is the meaning of the sentence “…in vivo model were performed in duplicate.”?
If the CFU counts and/or the histological slides were performed in duplicate this should be specified.
R/ It is a way to explaining that both the animal model and the CFU analysis were performed twice during the time of the study.
Results section
Table 1 is not fully readable and must be centered.
R/ Table 1 was redesigned in the manuscript
Line 287: Table 2 title should be modified as follows “Antifungal activity of antimicrobial peptides against different species/strains of Candida and Cryptococcus”
R/ Thanks for the detailed revision, the manuscript was modified
Line 290: In legend of Figure 1 the “parent” peptide Ox3-22 should be added.
R/ Information added to the manuscript
Figure 2: The label of Y axis should be “% Metabolic activity” instead of % Biofilm formation since no real data on biofilm formation have been collected.
R/ The chart in Figure 2 was modified considering the reviewer's suggestion.
The effect of the investigated peptides on C. albicans ATCC 10231 metabolic activity are not apparently related to the previously obtained MIC values. Moreover, amphotericin B, a fungicidal agent, has a lower effect even at high concentrations. Is there some possible explanation? What are the values of MIC and MFC for amphotericin B against C. albicans ATCC 10231?
R/: We agree with the reviewer's comments. However, Figure 2 shows a significant decrease in the metabolic activity of yeasts exposed to high concentrations of amphotericin B. This effect was possibly not so marked because this analysis was not performed on planktonic cells but on Candida biofilms, and it is known that biofilm hinders the penetration of antifungal drugs. The MIC value of AMB for C. albicans ATCC 10231 is 0,19 - 0,25 μg/mL. Please consider that normal antifungal susceptibility testing uses a yeast concentration of 1-5×103 cell/ml (CLSI manual), while biofilm experiments start with 2×106 cell/ml.
Lines 307-308: Since the effect of the investigated peptides on C. albicans morphogenesis has been evaluated without reference to quantitative data (see the previous comment on Materials and Methods Paragraph 2.7. Activity against morphogenesis of C. albicans) the statement “Yeasts treated with Act-6 and Act 8-20 showed a significant reduction in the formation of both hyphae and pseudohyphae compared to untreated control cells.” is not correct. Microscopic observations should be merely descriptive.
R/ Following the reviewer's recommendations, the expression "significant reduction" was removed from the manuscript because it was not a quantitative analysis.
Figure 3, panel A: While the with arrows surely indicate hyphae, the structures indicated by red arrows (although the image is not very clear) more probably refer to germinating yeast cells (germ tubes, the initial phase of hypha formation). Pseudohyphae are likely present (but cannot be clearly individuated) within the mass of fluorescent material in the upper part of the figure.
R/ According to the reviewer's comments, these may be germ tubes or pseudohyphae in formation.
Line 329: The sentence “Act-6 and Act 8-20 at both 25 and 6.25 μg/mL…” should be corrected in “Act-6 and Act 8-20 at 25 and 6.25 μg/mL respectively…”
R/ Sentence modified in the manuscript
Lines 353-356: The different results obtained in histopathological analysis of different organs (i.e. decrease in the fungal burden in the livers of mice treated with either Act-6 or Act 8-20 while there were no differences in the histopathological analysis of kidneys and spleens of untreated and treated mice), also considering the results obtained in the analysis of the organ fungal burden (i.e., decreasing in fungal burden after peptide treatment more relevant in kidneys and spleens where the fungal burden in untreated mice was higher), could be explained by the lack of quantitative data?
R/ The reviewer's concern is very pertinent. However, our histopathological analysis is not based on the count of fungal cells found, but rather on their presence or absence, comparing the organs of infected and treated animals with those of infected animals only. We reiterate that, using this staining technique for the histopathological analysis, we were unable to find yeast in the kidneys and spleens of any of the groups of mice analyzed. We clearly understand that, based on the results of the fungal burden assay of the same tissues, there was exposure to the fungus; most likely, the staining technique used did not allow us to corroborate these results.
Discussion section
Line 367: Based on the references 31 and 32, cited at the end, the sentence should be “Mammalian AMPs also possess antimicrobial capacity and…”
R/ Sentence modified in the manuscript
Line 386: The sentence “The three peptides demonstrated low hemolytic activity,” could be modified, since Act 8-20 showed higher cytotoxicity against red blood cells.
R/ Sentence modified in the manuscript
Lines 409-411: The morphological changes observed by TEM in some yeast cells treated with peptides cannot be directly related to membrane disruption, other mechanisms of action of antimicrobial peptides could cause similar cell alterations. For this reason, the last part of the sentence “… suggesting membrane disruption.” could be deleted.
R/ Sentence modified in the manuscript
Lines 411-413: A slight modification of the sentence is suggested for clarity “Other AMPs, like
PNR20 (100 μM) [28], and K40H, an immunoglobulin M-derived peptide [42], also induce morphological alterations, such as cytoplasm disintegration, vacuolation, and cell wall swelling.
R/ Sentence modified in the manuscript
Lines 425-426: The sentence “This contrasts with PNR20, which showed synergy with fluconazole in vivo for disseminated candidiasis [45].” seems unrelated with the previous statements. Deletion is suggested.
R/ Sentence deleted from the manuscript
Reference section
Lines 523-526: Reference 24 should be correctly reported by deleting the part “Guidelines for Diagnosing, Preventing and Managing Cryptococcal Disease among Adults, Adolescents and 524 Children Living with HIV; 3rd ed.;”
R/ Reference 24 was corrected
This manuscript is a resubmission of an earlier submission. The following is a list of the peer review reports and author responses from that submission.
Round 1
Reviewer 1 Report
What I consider should be checked are some conclusions reached by the authors that seem excessive to me for some reported results:
1. With what evidence do the authors assure that the reduction in metabolic activity is directly related to the decrease in the biofilm-forming capacity? (Discussion, paragraph 382-394).
2. Vesiculation is observed in the fungal cells and probably as part of the cell death process the membranes become permeabilized. In my opinion, it cannot be deduced from the electron micrographs presented that the membrane is permeabilized. For this, an assay would be needed with a fluorophore such as Sytox Green or Propidium Iodide, which cannot be internalized by the fungal cell unless the membrane has lost its continuity. (Discussion, paragraph 405-418)
M. Bacalum & M. Radu in “Cationic Antimicrobial Peptides Cytotoxicity on Mammalian Cells: An Analysis Using Therapeutic Index Integrative Concept” (International Journal of Peptide Research and Therapeutics Volume 21, pages 47–55, 2015) provided a unitary way to characterize the side effects of AMPs using a global concept of therapeutic index. I suggest the authors calculate an integrative therapeutic index such as this one as a parameter to characterize the AMPs cell selectivity. These results should be considered in the discussion section.
Title, I suggest a minor modification: Antimicrobial Peptides Act-6 and Act 8-20 Derived from Scarabaeidae Cecropins Exhibit Strong Antifungal Activity. I think the word new should be avoided from titles as one can read the article in a while and what now is new would be old.
Lines 279-280 Since the substitutions are based on the Ox3-22 sequence, in my opinion, it would be less confusing to name the changes in reverse as is usual for substitution mutations but also in the design of peptides (e.g. V15I instead of I15V). Additionally, Ala is missing in the substitution of position 23.
Author Response
Response to reviewer 1's comments
- With what evidence do the authors assure that the reduction in metabolic activity is directly related to the decrease in the biofilm-forming capacity? (Discussion, paragraph 382-394).
R/ Different studies indicate that metabolic activity is related to the early capacity for Biofilm formation (PMID: 29487851), which in this case was quantified by determining metabolic activity (PMID: 16332875), for this reason we add to the discussion that the metabolic activity is directly related to biofilm-forming capacity.
- Vesiculation is observed in the fungal cells and probably as part of the cell death process the membranes become permeabilized. In my opinion, it cannot be deduced from the electron micrographs presented that the membrane is permeabilized. For this, an assay would be needed with a fluorophore such as Sytox Green or Propidium Iodide, which cannot be internalized by the fungal cell unless the membrane has lost its continuity. (Discussion, paragraph 405-418)
R/ In accordance with the reviewer's comments, we specifically removed the membrane permeabilization issue from the discussion and only left the other induced morphological alterations.
- Bacalum & M. Radu in “Cationic Antimicrobial Peptides Cytotoxicity on Mammalian Cells: An Analysis Using Therapeutic Index Integrative Concept” (International Journal of Peptide Research and Therapeutics Volume 21, pages 47–55, 2015) provided a unitary way to characterize the side effects of AMPs using a global concept of therapeutic index. I suggest the authors calculate an integrative therapeutic index such as this one as a parameter to characterize the AMPs cell selectivity. These results should be considered in the discussion section.
R/ We are grateful with the reviewer for this comment. We found your study in the International Journal of Peptide Research and Therapeutics (Volume 21) very insightful. While a unitary TI certainly provides a useful metric for AMP cytotoxicity assessment, I propose that a more nuanced approach might be beneficial. Cellular Diversity: Different mammalian cell types exhibit varying sensitivities to AMPs. A global TI may not accurately reflect toxicity against specific cell lines that could be relevant for the AMP's intended therapeutic application.
Selective Targeting: The goal of AMP development is often to achieve selective toxicity towards pathogens while minimizing harm to host cells. A cell-type specific TI would better highlight AMPs with high selectivity against the cells involved in an infection.
Clinical Relevance: In clinical settings, understanding the potential impact of AMPs on specific healthy cell types is crucial for minimizing side effects. Cell-type specific TIs would provide more clinically relevant toxicity data.
4. Title, I suggest a minor modification: Antimicrobial Peptides Act-6 and Act 8-20 Derived from Scarabaeidae Cecropins Exhibit Strong Antifungal Activity. I think the word new should be avoided from titles as one can read the article in a while and what now is new would be old.
R/ We agree with the reviewer's comment; the change was made to the manuscript´ title
Reviewer 2 Report
The article deals with an important topic, which proposes alternative molecules for the control of fungal infections caused by species that have demonstrated resistance to conventional medicines. The work presents results that demonstrate the antifungal capacity of the molecules studied, in particular ACT 8-20, which had an effect at lower concentrations. The study is very relevant and will allow the design of new molecules inspired by this work.
Some corrections must be made such as:
Lines 22, 92, 187: the term "in vitro" should be written in italics
Lines 25, 96: the term "in vivo" must be written in italics
Lines 175, 201, 213, 226, 296, 306, 323: species of microorganisms must be written in italics
Item 2.3 (hemolytic activity assay) does not contain the ethics committee protocol number for studies involving human cells. This must be arranged for such publication.
Table 1 needs to be revised, for example, the amino acid sequence of each peptide must be configured on the same line
I suggest suppressing the results referring to the ACT6 + FCZ and ACT8-20 + FCZ combinations referring to Figure 5. These results may involve associations of different types (synergism, competition, etc.) that are not discussed in the article. Therefore, they should not be presented in this work.
Author Response
Response to reviewer 2 comments:
Lines 22, 92, 187: the term "in vitro" should be written in italics
R/ We agree with the reviewer, however, the editorial staff of JoF journal suggest us not to write the word in vitro and in vivo in italics.
Lines 175, 201, 213, 226, 296, 306, 323: species of microorganisms must be written in italics
R/ We agree with the reviewer, usually the name of the species should be written in Italic. However, the magazine's guidelines indicate that in the case of titles, all text must be in italics except the name of the species.
Item 2.3 (hemolytic activity assay) does not contain the ethics committee protocol number for studies involving human cells. This must be arranged for such publication.
R/ We appreciate the reviewer´ observation. In this case, the ethics committee approval was added in the materials and methods section where studies involved human cells.
Table 1 needs to be revised, for example, the amino acid sequence of each peptide must be configured on the same line.
R/ Corrections were made in table 1.
I suggest suppressing the results referring to the ACT6 + FCZ and ACT8-20 + FCZ combinations referring to Figure 5. These results may involve associations of different types (synergism, competition, etc.) that are not discussed in the article. Therefore, they should not be presented in this work.
R/ We appreciate the reviewer's comment. We decided to include in the discussion a section about the synergism between antimicrobial and antifungal peptides such as fluconazole, for this reason we prefer to leave the synergism results in Figure 5 (ACT6 + FCZ and ACT8-20 + FCZ).
Reviewer 3 Report
Dear authors,
Minor revisions are suggested before the publication of the manuscript
Line 63 I kindly suggest the inclusion of Cryptococcus grubii to your pathogenic list. Some papers reveal the relevance of this species, eg, DOI: 10.1016/j.ijantimicag.2024.107090
Line 71 Antimicrobial Peptides come not only from animals, some reviews about plant and microbe AMPs are available in the literature, eg. DOI: 10.3390/ph8040711, 10.3390/cimb45040239, 10.1111/j.1365-2672.2012.05338.x, and 10.1016/j.peptides.2005.03.007
Line 120 Characterization of the peptides (Act-6 and Act 8-20) is important since the structure and purity are the key in antifungal activity. Please add the HPLC chromatogram and MS-spectrometry analysis of your peptides.
Line 167 The emission and excitation values seem to be swapped. Check your values.
In the discussion section, a deeper analysis comparing your results (MIC, biofilm formation inhibition, etc.) with those obtained for other AMPs or even organic molecules tested as antifungal is needed to emphasize the relevance of your work, especially for Act 8-20.
Regards
Dear authors,
Minor revisions are suggested before the publication of the manuscript
Line 63 I kindly suggest the inclusion of Cryptococcus grubii to your pathogenic list. Some papers reveal the relevance of this species, eg, DOI: 10.1016/j.ijantimicag.2024.107090
Line 71 Antimicrobial Peptides come not only from animals, some reviews about plant and microbe AMPs are available in the literature, eg. DOI: 10.3390/ph8040711, 10.3390/cimb45040239, 10.1111/j.1365-2672.2012.05338.x, and 10.1016/j.peptides.2005.03.007
Line 120 Characterization of the peptides (Act-6 and Act 8-20) is important since the structure and purity are the key in antifungal activity. Please add the HPLC chromatogram and MS-spectrometry analysis of your peptides.
Line 167 The emission and excitation values seem to be swapped. Check your values.
In the discussion section, a deeper analysis comparing your results (MIC, biofilm formation inhibition, etc.) with those obtained for other AMPs or even organic molecules tested as antifungal is needed to emphasize the relevance of your work, especially for Act 8-20.
Regards
Author Response
Line 63 I kindly suggest the inclusion of Cryptococcus grubii to your pathogenic list. Some papers reveal the relevance of this species, eg, DOI: 10.1016/j.ijantimicag.2024.107090
R/ Thanks to the reviewer for this comment. C. neoformans var. grubii is indeed the main cause of cryptococcosis worldwide and it is usually referred to as C. neoformans. In fact, the strain of C. neoformans used in our study, H99, is precisely C. neoformans var. grubii (serotype A). As stated in line 63, we use the nomenclature "C. neoformans and C. gattii species complexes" to make reference to all varieties, molecular types and lineages, that have been described in these 2 species.
Antimicrobial Peptides come not only from animals, some reviews about plant and microbe AMPs are available in the literature.
R/ According to the reviewer's comment, this information was added to the manuscript.
Line 120 Characterization of the peptides (Act-6 and Act 8-20) is important since the structure and purity are the key in antifungal activity. Please add the HPLC chromatogram and MS-spectrometry analysis of your peptides.
R/ We consider that it is not relevant to show a figure with the result of the HPLC carried out by the outsourced company Peptide 2.0, responsible for the synthesis of the peptide and which is also widely recognized in the synthesis of biomolecules.
Line 167 The emission and excitation values seem to be swapped. Check your values.
R/ We appreciate the reviewer's comment; this information was modified into the manuscript.
In the discussion section, a deeper analysis comparing your results (MIC, biofilm formation inhibition, etc.) with those obtained for other AMPs or even organic molecules tested as antifungal is needed to emphasize the relevance of your work, especially for Act 8-20.
R/ According to the reviewer's comment, this information was added to the manuscript.
Reviewer 4 Report
The manuscript by Rodríguez et al. deals with the antifungal activity of two antimicrobial peptides obtained through modification of a previously described cecropin (Oxysterlin 3) derived from Oxysternon conspicillatum.
The manuscript fits in the consolidated line of research aimed at identifying peptides with antimicrobial activity, in particular antifungal activity, to overcome the problems of toxicity and spread of resistance related to the existing drugs.
Although the novelty of the contribution is scarce, the addition of new molecules to the potential candidates for the development of antifungals is always welcome. The presented results are preliminary to deeper investigations on the mechanism of action of these molecules and, possibly, a further optimization to balance the efficacy and the toxicity.
see attachment

Author Response
Some errors in the text should be corrected (for example, Line 94: short peptides, instead of peptide, or Line 178: SDA, instead of SAD). Moreover, microorganism names in references 20, 30, 36, 37, and 39 should be in italics.
R/ We appreciate the reviewer's comment; this information was modified into the manuscript.
A first comment relates to the assessment (title, abstract line 27, and discussion line 438) of the “Strong Antifungal Activity” of the investigated peptides. The suggestion is to substitute with “differential antifungal activity” to underline that the results were not the same for the different species and strains investigated. Furthermore, since these differences cannot be explained without a deeper investigation into the mechanisms of action of the peptides (not addressed in this manuscript), a brief comment on this point could be added in the discussion section.
R/ We thank the reviewer's comment; the manuscript was modified.
A first preliminary contribution to the study of the mechanism of action could rely on the establishment of the possible direct fungicidal activity of the peptides. Why the minimal fungicidal concentration (if any) has not been evaluated together with the MIC?
R/ We appreciate the reviewer's suggestion, it is a very valid comment and we will take it into account for future trials with antimicrobial peptides. Herein the trailing growth is defined as residual growth at drug concentrations higher than the MIC values, was not observed in our experiments, we did not carry out fungicidal activity assays. However, we will consider this in further experiments testing AMPs..
Line 38: it would be better to write “…skin and mucosal infections…” instead of “…skin infections at the mucosal level…”
R/ The manuscript was modified.
Line 58: reference 7 could be substituted by a more recent paper
R/ The manuscript was modified.
Line 60: reference 8 could be substituted by a more general paper and the same paper now cited as ref. 8 possibly moved to the next paragraph (substituting ref. 11).
R/ The manuscript was modified.
Lines 74: reference 13 could be substituted by a more recent paper, while reference 14, related to anticancer activity of antimicrobial peptides, is not relevant and could be deleted.
R/ The manuscript was modified.
Line 87 and line 95: It seems that Reference 18, cited in line 87, and reference 20, cited in line 95, have been exchanged. Please check and correct. Note that the number should be changed also in line 112 (Materials and Methods section) where the correct reference is cited (now as ref.18). Please check also the correct citation of the same papers in the discussion section.
R/ The manuscript was modified.
Paragraph 2.3. Assessment of the hemolytic activity of the AMPs. In line 138 it is stated: “The plate was then incubated at 37 °C for 18 h…” Is this correct? To my knowledge, incubation for the evaluation of the hemolytic activity of peptides is generally 30 minutes, 1 hour or 2 hours. In reference 21, cited by the authors (line 125: The hemolytic activity … was evaluated … as reported previously [21]) the incubation period was 1 h. Please provide an explanation.
R/ According to the reviewer's comment, in this experiment the plate was incubated at 37 °C for 18 h. We consider relevant to incubate the plate for more hours to better demonstrate the possible hemolytic effect of the peptides studied. However, this effect was low during the time analyzed.
Paragraph 2.4. Evaluation of the cytotoxic activity of the AMPs. In line 149 is again cited reference 21. Since no evaluation of cytotoxic activity is mentioned in this paper, please check and provide the correct reference that is now lacking.
R/ The manuscript was modified.
Line 138: reference 21 should be cited instead of 20.
R/ The manuscript was modified.
Lines 195-196: “Two AMPs with the best antifungal activity were selected for further experiments.” What’s the meaning of this sentence? The entire manuscript refers to the two peptides Act-6 and Act 8-20 designed starting from the template constituted by the residues 1-22 of the peptide Oxysterlin 3. No other peptide is mentioned.
R/ The manuscript was modified.
Paragraph 2.7. Activity of AMPs against pseudohyphae formation by C. albicans. This is, in my opinion, a very critical issue. Why the authors refer to pseudohyphae formation? The images presented in the results show the inhibition of C. albicans germination and the formation of hyphae (not pseudohypahe). (see below the comment on Results and Discussion sections).
R/ We appreciate the comment of the reviewer. Regarding the term pseudohyphae specifically in the species C. albicans has been widely used in the literature (PMID: 30447520, PMID: 36656405, PMID: 23242994). Formation of these pseudohyphae could lead to the true hyphae formation. In our study we clearly observed, how antimicrobial peptides have the ability to inhibit the formation of pseudohyphae; an important virulence factor of C. albicans.
Line 267: “… in vivo model were performed in duplicate.” What’s the meaning of this sentence? Did the authors perform in duplicate the assays for the determination of the fungal burden or the histological slides (or both)? Please clarify.
R/ Certainly, both the in vivo model and the histological slides were performed twice separately
First Paragraph. It would be useful for the reader to have a paragraph dedicated to the features of the selected peptides, for example “3.1 Features of the designed AMPs” and then a second Paragraph 3.2. Act-6 and Act 8-20 antifungal activity against different species of Candida and Cryptococcus.
R/ The information requested was added in section 3.1.
Lines 286-287: “the peptide Act 8-20 showed the best antifungal activity” this statement is not true for all the investigated strains. Moreover, Act 8-20 showed a relevant increase in hemolytic and cytotoxic activity (not a surprising achievement due to the specific modifications introduced in this molecules). This issue should be briefly discussed in discussion section (modifying the text in lines 374-379)
R/ Table 2 shows the MICs results of the antimicrobial peptides studied. This table shows that the Act 8-20 peptide has greater antifungal activity compared to the other peptides. Only, in the case of the species C. parapsilosis ATCC 22019, Act8-20 was not as efficient, since it presented a MIC of 25 μg/mL, while the peptides Act-6 and Ox3-22 inhibited the growth of this species in concentrations of 3.12 and 0.39 μg/mL, respectively.
The data of decrease of metabolic activity shown in figure 2 deserve some comments: while peptide Act 8-20 at the highest concentrations completely abolish the metabolic activity of the investigate strain, C. albicans ATCC 10231, amphotericin B, a fungicidal antifungal, has a lower effect even at very high concentrations. What are the values of MIC and MFC for amphotericin B against C. albicans ATCC 10231?
R/ We agree with the reviewer's comments. However, Figure 2 shows a significant decrease in the metabolic activity of yeasts exposed to high concentrations of amphotericin B and antimicrobial peptides.
Line 301 (Figure 2 legend): Why Act-6 and Act 8-20 are indicated as antimicrobial defense peptides? These are not natural peptides, so the adjective defense is not appropriate.
R/ According to the reviewer's comment, this adjective was corrected in the indicated places.
Paragraph 3.3 Act-6 and Act 8-20 have the ability to inhibit pseudohyphae formation in C. albicans. See the previous comments on Paragraph 2.7. It should be written hyphae formation and not pseudohyphae formation. Also, in the legend of figure 3, it should be: “White arrows indicate hyphae” and NOT pseudohyphae.
R/ Comment answered above (Paragraph 2.7), as indicate by the reviewer.
Reviewer 5 Report
Authors evaluated the antifungal activity of designed short AMPs, Act-6 and Act 8-20, against pathogenic yeasts and their effects on biofilm formation, pseudohyphae formation, and in vivo activity against Candida albicans in a murine model. Both peptides showed strong antifungal activity, inhibited biofilm and pseudohyphae formation, induced morphological changes in C. albicans, and significantly reduced fungal load in mice. The authors' explanation of the results are insufficient, and there are additional concerns that require clarification from the authors.
The authors are advised to introduce Ox3-22 peptide in the introduction section for better contextualization.
In the legend of Figure 2, it says metabolic activity, but the figure was plotted using OD492. Additionally, it is recommended to include Ox3-22 in the analysis? Why does Amphotericin B fail to control Candida albicans ATCC 10231 in Figure 2?
Authors include brightfield images in Figure 3 for better clarity and comprehensive presentation of their findings.
The statistical tests utilized by the authors in Figure 5 are not explicitly stated.
The results provided are quite concise and lack thorough explanation. Authors should add more metrics to explain the data.
The authors noted a notable reduction in the fungal load within the livers of treated animals. We’re scoring methods employed to quantify this reduction, or were biochemical assays conducted to assess liver damage markers?
Author Response
The authors are advised to introduce Ox3-22 peptide in the introduction section for better contextualization.
R/ The detailed information about Ox3-22 peptide is on lines 122 to 128.
In the legend of Figure 2, it says metabolic activity, but the figure was plotted using OD492. Additionally, it is recommended to include Ox3-22 in the analysis? Why does Amphotericin B fail to control Candida albicans ATCC 10231 in Figure 2?
R/ We appreciate the reviewer's comments. The legend of Figure 2 was corrected, we did not add data on the Ox3-22 peptide in any figure of the manuscript because new peptide sequences were obtained, which are the main objective of the study. On the other hand, figure 2 shows that there is a significant decrease in the growth of C. albicans 10231 when exposed to Amphotericin B.
The statistical tests utilized by the authors in Figure 5 are not explicitly stated.
R/ In lines 364 and 365 we specified that the significant difference observed in Figure 5 was obtained when comparing the kidneys of untreated infected animals versus those from infected and treated animals with antimicrobial peptides associated or not with Fluconazole.
The authors noted a notable reduction in the fungal load within the livers of treated animals. We’re scoring methods employed to quantify this reduction, or were biochemical assays conducted to assess liver damage markers?
R/ To quantify the decrease in the liver fungal burden of the animals treated with the antimicrobial peptides, the counting colony-forming units’ methodology was used. No markers of liver damage were utilized in this study.
Round 2
Reviewer 1 Report
Hemolytic activity and cytotoxic activity of the AMPs were evaluated, and therapeutic index (TI) was calculated for the hemolytic and cytotoxic concentrations of the peptides Ox3-22, Act-6, and Act 8-20; results are shown in Table 2 and Figure 1 but the relevance of these results were not discussed in Discussion section.
No comments.
Reviewer 4 Report
The authors have NOT addressed satisfactorily most of the specific issues raised (see details in attachment)
see attachment
